# The Complex Network between Inflammation and Colorectal Cancer: A Systematic Review of the Literature

**DOI:** 10.3390/cancers13246237

**Published:** 2021-12-12

**Authors:** Rossana Percario, Paolo Panaccio, Fabio Francesco di Mola, Tommaso Grottola, Pierluigi Di Sebastiano

**Affiliations:** 1General Surgery Unit, “F. Renzetti” Hospital, 66034 Lanciano, Italy; rossana.percario@asl2abruzzo.it (R.P.); paolo.panaccio@asl2abruzzo.it (P.P.); 2Division of Surgical Oncology, Clinica Pierangeli, 65124 Pescara, Italy; fabio.dimola@grupposynergo.com (F.F.d.M.); tommaso.grottola@grupposynergo.com (T.G.); 3Department of Medical, Oral and Biotechnological Sciences, University D’Annunzio Medical School, 66100 Chieti, Italy

**Keywords:** colorectal cancer, inflammatory bowel diseases, chronic inflammation, molecular pathways, tumorigenesis, PRISMA statement guidelines

## Abstract

**Simple Summary:**

Colorectal cancer is one of the most diffuse malignant pathologies, and many factors are involved in its genesis. Among these factors, inflammation plays an important role. Many molecules are involved in inflammation processes and are linked in different pathways, either in the sense of pro-tumorigenesis or anti-tumorigenic action. This review was conducted with the aim to review in a single paper the majority of actual knowledge in the literature and to comprehend inflammation patterns for better clinical and surgical management of patients.

**Abstract:**

Background: colorectal cancer (CRC) has a multifactorial etiology which comprises microbiota, genetic predisposition, diet, environmental factors, and last but not least, a substantial contribution by inflammation. The aim of this study is to conduct a systematic review of the literature regarding the strong link between inflammation and colorectal cancer. Methods: A systematic review of the literature on PubMed (Medline), Scopus, Cochrane and EMBase databases was performed, following the PRISMA 2020 guidelines. Each paper was reviewed by two groups of researchers in a single-blind format by using a pre-planned Microsoft^©^ Excel^®^ grid. Results: Using automated research filters, 14,566 studies were included, but 1% was found significant by the reviewers. Seventy pathways of inflammation were described in the sequence of inflammation-carcinogenesis, and anti-tumorigenic molecules were also found. Conclusion: several studies suggest a strong role of inflammation in the tumorigenesis of colorectal cancer through different pathways: this may have a diagnostic and clinical role and also therapeutic purpose in preventing carcinogenesis by treating inflammation. In vitro tests support this theory, even if many other clinical trials are necessary. The present paper was registered in the OpenScience Framework registry (Identifier: DOI 10.17605/OSF.IO/2KG7T).

## 1. Introduction

According to a World Health Organization evaluation, cancer is the second leading cause of death globally, accounting for 9.6 million deaths in 2018 (with a ratio of 1:6 of overall deaths) [1]. In 2020, in 27 countries of the European Union, colorectal cancer accounted for 12.7% of all new cancer diagnoses and 12.4% of all deaths due to cancer being the second most frequently occurring cancer (after breast cancer) and the second greatest cause of cancer death (after lung cancer). An increased number of new diagnoses of colorectal cancer is probably also due to the augmented awareness of the population on cancer matters and, consequently, the increased number of people who join screening programs (i.e., fecal occult blood testing and colonoscopy) which allow an early diagnosis and better prognosis [2].

Pre-clinical and clinical studies aim to define risk factors, pathological history and, in some cases, to try to find a preventive and/or tailored treatment. Microbiota, genetic predisposition, diet, environmental factors and inflammation are described as pathogenetic factors, and they may start the carcinogenetic process with various combinations.

Here the authors focused their attention on a particular subgroup of CRC: Colitis-Associated Carcinoma (CAC).

Although there is a day-by-day evolution of knowledge in the pre-clinical and clinical field, here, the authors attempt to review all known inflammation factors, dividing them into three groups: pro-tumorigenic, anti-tumorigenic and hybrid action, and then putting them into organic pathways.

In this complex pathology, relationships between gastroenterologists, physicians of internal medicine, surgeons, oncologists and pathologists must follow a very strict multidisciplinary approach in which everyone knows the evolution of the pathology from first signs of inflammation to the last evolution in Colitis-Associated Carcinoma (CAC). This approach aims to assure the patient of the best diagnostic pathway and the best treatment possible, keeping in mind that CAC is different from sporadic Colorectal Cancer.

## 2. Materials and Methods

The research team includes five surgeons with experience in colorectal cancer surgery who are part of a multidisciplinary board for colorectal cancer therapy.

The research work started with the choice to adhere to the Preferred Reporting Items for Systematic Reviews and Meta-Analyses (PRISMA^®^) Statement Guidelines 2020 [3].

The first step was the preparation of the research protocol by two of the researchers: the protocol was submitted to the attention of the other three researchers with the aim to avoid procedural biases.

The team defined the keywords (inflammation, colon cancer, rectal cancer, colorectal cancer) and databases (PubMed (Medline), Scopus, Cochrane and EMBase). The choice to have a keyword string so wide was made to have the greatest number of papers possible in the beginning.

Search Results were filtered by English language, type of publication (full-text article), subject of research (human only) and publication date (from 1 January 2000 to 31 January 2021).

A Microsoft^®^ Excel^®^ database was prepared, which contained bibliographical information (title of journal, authors, DOI, pages, year of publication, etc.) and eight parameters of evaluation. For each of the parameters, a scoring system from 1 to 5 was assigned in increasing order of significance. The parameters were: quality of title—i.e., relevancy to the main theme of the research; quality of abstract—i.e., clear target, clearly planned and exposed, relevance of keywords; quality of the full text—i.e., clear exposure of study design; characteristics of population (for this item the judgment was homogeneous or not); size of population (for this item the judgment was statistically not representative, low/medium/high statistically representative); analytics procedure (declared/not declared); bias (declared/not declared); intracellular pathway (yes or no). The last column of the Excel^®^ database was reserved for brief notations, in particular, about causes of exclusion, with the aim to have an immediate reminder of motivations for low scores. Table 1 shows the analyzed criteria and scoring system.

A subgroup of two authors conducted the first sorting of papers. The first scoring evaluated title, abstract and keywords and then rated them using the previously mentioned scoring system.

Papers with a mean score of 4 and 5 were directly admitted, papers with a score of 1 or 2 were directly eliminated, papers with 3 were put on a separate waiting list for a second review.

After this, as in a randomized, single-blind study, 20% of all considered papers (both included and excluded) were brought to the attention of the second subgroup of the team, who proceeded with the same rules of evaluation as the first group, with the aim to avoid bias of selection of papers. Researchers fixed, as acceptable, an agreement of at least 90% to be able to proceed further.

The first two authors proceeded for the second sorting of papers by considering full text with no consideration of the previous scoring; therefore, a new Microsoft^®^ Excel^®^ database sheet was used, containing bibliographical indications alone. The sorting was made, always using the same scoring method from 1 to 5.

Even in this step, with the same modality, the second subgroup of authors was involved.

The team proposed the two-subgroup model with the aim of reducing bias in the selection phase.

## 3. Results

### 3.1. Study Selection

As shown in Figure 1, our initial search identified 73,328 papers among Pub-MED MEDLINE and Elsevier’s databases. By applying the filters previously mentioned and removing duplicates, 14,527 papers were left to process.

After the first sorting, 354 papers were found eligible (155 with scores 4 and 5 and 199 with score 3), while, after the second sorting, 153 papers were considered suitable for the present review.

The concordance between the two groups of authors was 93% for the first step (title, abstract and keywords) and 97% for the second step (full text). All the articles that explain intracellular pathways were deliberately excluded, according to the aim of the study.

The main causes of exclusion of papers were: total score lower than 2 for 127 papers (Reason 1 on PRISMA flowchart), the presence of murine/animal subjects not declared in the title, keywords or abstract for 32 papers (Reason 2), the coexistence in the same paper of other tumors except CRC for which inflammation drives carcinogenesis, for example, lung, prostate, etc., for 42 papers (Reason 3).

### 3.2. Molecular and Cellular Subject

Nowadays, it is well known that there is complex crosstalk of different pathways in terms of enhancement or inhibition. Lu et al. reported this in a flow-diagram obtained by considering each molecule as a Boolean variable [4].

The factors involved in inflammation can be divided into three groups: (1) factors whose inactivation reduce inflammation and, therefore, the possibility of developing CAC; (2) those whose inactivation may exacerbate inflammation and possibly lead to the development of CAC; (3) factors with a more complicated role, e.g., may increase the inflammation, but this may instead have a protective role against CAC [5]. Table 2 presents the role of each factor.

Here, we present the abovementioned factors with their role in inflammation.

#### 3.2.1. IL-6

IL-6 plays a role in inflammation, immune regulation and oncogenesis [6]. It is produced by monocytes/macrophages, B and T lymphocytes and by epithelial and malignant cells under stimulation of PGE2 and TGF-β, indirectly via NF-kB (NF-kB, in this case, is activated by LPS and IL-1β) and under autocrine stimulus via the JAK2/STAT3 pathway. High levels of IL-6 in serum increase the risk of developing colorectal cancer, and this is considered an independent negative prognostic survival factor [7,8]. Binding this receptor (IL-6R), IL-6 activates the JAK2/STAT3 pathway, thus, inducing the activity of MetalloProteinase (MMP-2, MMP-7 and MMP-9) in the degradation of the extracellular matrix (ECM) and the transcription of vascular endothelial growth factor (VEGF) through hypoxia-inducible factor-1 (HIF-1) [6,8]. IL-6 also acts in the up-regulation of c-myc activity, which, in turn, promotes the production of a ribosomal protein in p53 degradation; in this way, DNA damage induced by ROS cannot be repaired due to the loss of function of the p53 gene [9]. Il-6, in collaboration with TGF-β, triggers antigen-presenting cells (APCs), recruits and activates Th17 cells and silences TReg cells and is associated with poor disease-free survival [8,10].

#### 3.2.2. IL-11

IL-11 is a part of the IL-6 family; it is produced by the same cells and under the same stimuli (except hypoxia-inducible factor, which impacts only IL-11 production) as IL-6 and shares 22% of its protein sequence [8]. Despite that, IL-11 has a greater gastrointestinal tropism than IL-6 [11] and has an anti-inflammatory role in contrast with IL-6. It has thrombopoietic and erythropoietic activity and drives the production of IgG by B cells [8].

#### 3.2.3. IL-23

Considered a member of the IL-12 family, IL-23 is produced by APCs in an inflammatory environment. Its expression correlates with TNFα, IL-6, IL-17, IL-22 IFN-γ and VEGF [7]. It acts in immune-mediated activity in the mucosal damages in IBD [6] through Th17 activity stabilization and TReg activity suppression [7]. Regarding Th17 stabilization, IL-23 is the only inductor of memory T cell proliferation and specialization in Th17 that produces IL-17 and IL-22; the link between IL-23 and Th17 is necessary for the survival of the latter [12].

#### 3.2.4. Chemokines CXCL8, CCL20, CCL2, CCL5

Their role in inflammation is to modify immune response, facilitating the inflow of inflammatory cells and decreasing the integrity of barriers. In particular, CXCL8 is increased in inflammation, while CRC is increased through CXCR1 and is involved in cells transition from normal to carcinoma [6]. CXCR2 promotes chronic inflammation and tumorigenesis and is necessary for in situ recruitment of myeloid-derived suppressor cells (MDSCs) [13]. CXCL3 expression for myeloid cells is regulated by K-Ras and enhances the production of cytokines.

#### 3.2.5. NF-kB

NF-kB has a role in modulation inflammation, stimulation of cancer cell proliferation and prolongation of their survival by blocking the activity of anti-apoptotic genes [6]. It acts as an antagonist of the p53 oncosuppressor gene [14] and as an activator of IL-6, IL-11, IL-22, HGF and EGF. In some actions, NF-kB and STAT3 overlap their activities [5]. In CAC, epithelial NF-kB is responsible for the number of cancer foci, while macrophage NF-kB is responsible for tumor size [15]. Moreover, its presence denotes late tumor stage, chemoresistance and poor outcome. NF-kB is involved in two different pathways: the canonical pathway through TNFα and IL-1 and non-canonical through TNF superfamily and receptor-activated NF-kB ligand (RANKL) [16].

#### 3.2.6. IkB

IkB acts as an inhibitor of NF-kB in response to pro-inflammatory cytokines and DNA damage. Its deletion impacts tumor size via IL-6 [17].

#### 3.2.7. ROS

They start the IL-1, IL-6, TNFα and IFNγ cascade, disrupting tight junctions between epithelial cells [6] and activating Nf-kB [10]. ROS are produced by activated macrophages but also by pre-malignant cells via TNFα and IL-1 stimulation [5].

#### 3.2.8. RNS

Reactive nitrogen species act as regulators of tumor blood support by regulating IL-8 and VEGF production with a direct vasodilatation effect. The release is under control of the JAK2/STAT3 pathway, which is started by IL-22 and IL-6, while NF-kB is, in turn, activated by IL-1β, TNFα and STAT1 even if NF-kB is not essential as a STAT3 pathway [18].

#### 3.2.9. TNFα

The role of tumor necrosis factor-α is in the activation of immune cells and subsequent carcinogenic activity by ERK, NF-kB and PI3K-AKT pathways resulting in increased levels of β-catenin [5,19,20]. Via NF-kB, TNFα reduces MUC2 activity with the effect of reducing the production of the mucin layer, which exposes the underlying cell to damage [21]. TNFα is produced by resident cells in the mucosa [17] as monocytes/macrophages, mast cells, B and T-lymphocytes, NK, neutrophils and endothelial cells [7].

#### 3.2.10. COX-1

Cycloxygenase-1 is involved in maintaining basal levels of prostanoids, which is important for normal tissue homeostasis [13].

#### 3.2.11. COX-2

Absent in normal intestinal cells, cycloxygenase-2 expression is regulated by pro-inflammatory cytokines and is elevated in CRC [13]. The levels of COX-2 are elevated in more than 90% of cancer and in more than 50% of adenomas, suggesting a fundamental role in carcinogenesis [22].

#### 3.2.12. PGE2

In physiological conditions, prostaglandin E2 supervises correct restitutio ad integrum of the epithelial layer after a harmful condition [22]. It accelerates tumor growth by silencing tumor suppressors and DNA repair genes, influences the activity of immune and endothelial cells via CXCL1 and CXCL2 [13] and inactivates ERK, NF-kB and PI3K-AKT pathways resulting in increased levels of β-catenin signaling [5]. PGE2 also stimulates the release of amphiregulin, which is one of the ligands of EGFR: its role is to amplify cell proliferation and induction of growth factors [15,22].

#### 3.2.13. IL-37

IL-37 is an immunosuppressive and anti-tumorigenic cytokine. It acts in suppressing the proliferation of epithelial cells via β-catenin, promotes cancer cells apoptosis and reduces the number of MSC. Depending on its localization in the tumor mass, IL-37 acts in different manners: high levels in peritumoral tissues, low levels in the tumor core, where it suppresses the production of IL-6 and β-catenin. It has been demonstrated that a lack of IL-37 may indicate a poor prognosis and metastatic spread [23].

#### 3.2.14. MDSCs

MDSCs are immature myeloid cells whose levels have a positive correlation with stage and metastatic spread. Called in situ by inflammation, MDSCs suppress CD8+T cell activation, proliferation, trafficking and activity. NK is also inhibited [13].

#### 3.2.15. IL-1

IL-1 is secreted by lamina propria, DC and macrophages, jointly with IL-8, and is regulated by TLR and NF-kB signaling. IL-1 manifests pleiotropic effects in relation to the disease phase. Binding IL-1Ra (antagonist) IL-1 inhibits pro-inflammatory actions [7], while binding IL-1R activates pro-inflammatory pathways [24].

#### 3.2.16. IL-17

The role of IL-17 is in the innate and adaptive immune response to pathogens. IL-17 is secreted by a particular class of CD4+T helper cells, named Th17, due to their ability to produce IL-17 [7]. IL-17 is also secreted by NK, T CD8+, eosinophils and macrophages [25]. It has been considered protective against colitis but with a pro-tumorigenic effect. Its level correlates negatively with prognosis [7]. Due to the high importance of IL-17 activity in tumorigenesis and anti-tumorigenesis, and for a better comprehension of the involved pathways, they are considered separately:IL-17 stimulates IL-6, which in turn activates the JAK2/STAT3 pathway for controlling CXCL1, COX-2 and IL-1β, leading to the transition from immature to tumorigenic myeloid cells and upregulating themselves [25];IL-17 stimulates NF-kB to push up glycolysis and give energetic sources to tumor cells [25];IL-17 activates colorectal adenocarcinoma cell lines with TNFα in production of HIF-1a (which activates c-myc pro-oncogene) and in the production of factors which allow survival and proliferation of tumor cells themselves [25];IL-17, in addition to IL-23 from tumor-associated myeloid cells, activates the STAT3 pathway, which reduces quantities of T CD8+ cells and stimulates TReg immunosuppressive action [25];IL-17 promotes the production of VEGF from endothelial cells [25];Il-17, when present in early phase tumors, has an anti-tumoral effect. IL-17 recalls tumor-infiltrating neutrophils in tumor sites for the release of myeloperoxidase and hydroperoxide, recruits NK cell numbers and activates (by facilitating bindings between NK and tumor cells and augmenting secretion of granzyme and perforin), lymphocytes and dendritic cells. IL-17 also activates the production of IL-12 for T cytotoxic activity while suppressing the production of IL-10 and IL-13 (protumorigenic interleukins) [25].

#### 3.2.17. Th17

They are differentiated from others CD4+Th cells thanks to TGFβ and IL-6 (STAT3 pathway). Their functions are amplified thanks to IL-21 and stabilized thanks to IL-23 [7]. Th17 also release IL-21, IL-22 and GM-CSF [25].

#### 3.2.18. IL-22

IL-22 is a member of the IL-10 family and is secreted by Th17 and NK cells. Its overexpression reduces the IBD gravity score [7]. It starts JAK2/STAT3 for production of NOS as IL-6 and STAT1 pathways and is the only cytokine among the complex panorama of mediators, which is incapable of mediating information between leukocytes instead of transmitting information from leukocytes to the rest of the cellular world (such as fibroblasts and tumor cells). IL-22 also induces the production of IL-8, MMP, SOCS, IL-10 [18].

#### 3.2.19. Epithelial Cells

They secrete cytokines and chemokines and are also a target by modifying activity and survival programs [7].

#### 3.2.20. TLR

Toll-like receptor is expressed at low levels in normal conditions and highly expressed in inflamed tissues, with the aim to correct intestinal barrier integrity and supervise wound healing programs [7,15]. When chronically activated by LPS, TLR4 is a potent inducer of COX-2 for the production of PGE2. In other words, the same receptor has an anti- and pro-tumorigenic role. All of the TLR family members have the potential to induce dysplasia, but TLR4 has the predominant role—if there is no TLR4, the tumor does not occur.

Anti-tumorigenic role: DCs fight against tumor cells. DCs are activated by TLR and act through antigen presentation, T cell activation and direct cytotoxic effects. Activation of TLRs on DCs regulates T cell activation not only via the class II major histocompatibility complex but also through TLR-induced signals in DCs that block the suppressive effect of regulatory T cells in an IL-6-dependent manner. TLR8 activation inhibits TReg functions, supporting anti-tumoral immunity [26].

Pro-tumorigenic role: TLR is able to send its pro-tumorigenic signals in tumor cells or in TME by enhancing NF-kB, which, in turn, activates its cascade made up of IL-1β, TNF-α, IL-6 [26].

#### 3.2.21. Stromal Cells (Fibroblasts and Myofibroblasts)

In physiological conditions, they are located in lamina propria and act in intestinal organogenesis, proliferation, differentiation in epithelial colonic cells, wound healing [22], maintaining structural and functional operativity of ECM by depositing ECM constituents and by degrading them via MMPs [7]. Beyond this structural activity, fibroblasts are also capable of secreting HGF, FGF, PDGF, TGF and stem cell factors. In normal conditions, fibroblasts have COX-1 expression [22]. Chemokines and tissue resident cells (such as epithelial, endothelial and mesenchymal cells) recruit fibroblasts in the tumor site (cancer-associated fibroblasts, CAFs), which will be transformed into myofibroblasts [27] with COX-2 expression; this makes myofibroblasts particularly respondent to PGE2 [22]. Thanks to IL-6 autocrine and paracrine stimulus, CAFs upregulate VEGF, activate the JAK2/STAT3 pathway, express MCP-1, CCL11/8/20, CXCL5 and IL-9 for cancer cell invasion, growth and survival. Through IL-6 and -8, CAFs promote S100A8/A9 activation in immature monocytes to allow differentiation in pro-tumorigenic macrophages [7]. Stromal cells are almost 60–90% of the cancer mass, and fibroblasts are the most represented cells. Cancer stroma fibroblasts secrete a greater quantity of IL-6 than other cancer cells and have a particular protein, aSMA protein, which has an effect on tumor growth [27].

#### 3.2.22. VEGF

Role in wound healing processes and in neo-angiogenesis in tumor sites [7].

#### 3.2.23. IL-8

Produced by N2 neutrophils thanks to autocrine signaling. IL-8 is responsible for the proliferation and survival of cancer cells, angiogenesis and tumor infiltration, regulating the actions of macrophages, neutrophils, endothelial cells, TRegs and cancer cells. IL-8 acts through AKT, MAPK, STAT-3 intracellular pathways, ERK extracellular pathway and SNAIL, which is a signal that leads to conversion from epithelial to mesenchymal cells necessary for tumor progression [10,20]. Moreover, IL-8 upregulates serine-proteases of ECM, which in turn, are involved in the disruption of the integrity barrier [10].

#### 3.2.24. IL-1β

IL-1β is secreted by macrophages and acts in promoting cell survival. In particular, IL-1β extends the survival of endothelial cells by stimulating the production of VEGF by vascular smooth muscle cells [7] and contributes to cancer invasiveness by activating STAT3 [28]. Another pathway is the induction of the WNT cascade that determines the growth of tumor cells [29].

#### 3.2.25. MSCs

Mesenchymal stem cells are derived from bone marrow or from adipose tissue near the tumor site [20]. In general, MSCs are protective against CAC by reducing inflammation and pro-tumorigenic in the case of sporadic colon carcinoma. MSC promotes T-cell apoptosis through Fas/FasL and increases the number of TRegs. MSC secretes MCP-1, which acts in calling T cells and in determining their apoptosis. The debris of this process increases the induction of TGFβ in macrophages. Increased TGFβ levels promote cell cycle arrest in the G1 phase [7]. MSCs also express COX-2 and migrate to the epithelial layer during inflammation for local production and releasing of PGE2 [15].

#### 3.2.26. IFNγ

Interferon-γ activates the PD-L1-PD1 pathway for T CD8+ and T CD4+ exhaustion [14].

#### 3.2.27. STAT

Signal transducers and activators of transcription are a family that acts as a controller of T cell maturation. STAT 1 and 4 are involved with Th1 cell differentiation for the effect of IL-12 and IFN-γ, STAT 6 is involved with Th2 cell differentiation thanks to IL-4 stimulus, while STAT3 is responsible for Th17 differentiation under IL-6 stimulus [30].

#### 3.2.28. STAT3

When activated by cytokines, such as IL-22 (from T and NK cells), in normal conditions, it has an important wound-healing role. Meanwhile, in inflammation and after IL-6 stimulation, STAT3 has an anti-apoptotic role [31] and maintenance of the inflammatory state thanks to Th17 and the suppression of TReg. In tumorigenesis, STAT3 acts as an activator for anti-apoptotic genes (such as Bcl-2), as a controller of the cell cycle via cyclin D1 and c-myc (which block passage from the G0 to G1 phase of the cellular cycle) and as a promoter of production of VEGF and bFGF. In some actions, STAT3 and NF-kB overlap their activities [5]. Another pathway necessary for cell recruitment in inflamed fields involves S1P, which gives stimulus to macrophages and dendritic cells to produce IL-6, generating the IL-6/STAT3/S1P/IL-6 self-activating loop. A small portion of STAT3 resides in mitochondria, with housekeeping relevance [31].

#### 3.2.29. IL-21

IL-21 is produced by T CD4+ and NK cells and acts as an activator of them in an antitumor sense by augmenting their cytotoxicity [30].

#### 3.2.30. TGF-β

Tissue growth factor β has a dual and contrasting functional role in tumorigenesis phases. In fact, it acts as an antitumor cytokine in the early stages by stopping uncontrolled cellular proliferation and suppressing pro-inflammatory and protumorigenic cytokines. In the late stages of tumorigenesis, TGF-β acts as a protumorigenic cytokine by suppressing the antitumor immune response and allowing transition from epithelial to mesenchymal cells [10]. Its isoform, TGF-1β, is expressed on the TReg cell surface [32].

#### 3.2.31. IL-10

Like TGF-β, IL-10 has a dual role in function depending on the stage of tumorigenesis [24].

#### 3.2.32. MMP2 and 9

MMPs are serine proteases produced by activated macrophages. Their role is essential in the invasiveness of tumor cells by degrading collagen IV, which is the main component of basal membranes [10].

#### 3.2.33. NOD2

Nucleotide-binding oligomerization domain-containing protein 2 acts in the activation of NF-kB and MAPK pathways, following the bacteria muramyl dipeptide (MDP) signal. NOD2 also functions in the downregulation of TLR [24].

#### 3.2.34. βcatenin

It is a protein member of the Wnt/β-catenin pathway, which is activated by TNFα, HGF, PDGF and FGF19. This pathway determines the cytoplasmic accumulation of β-catenin, which translocates into the nucleus and stimulates pro-oncogenic genes [33].

#### 3.2.35. T Cells

The presence of Th1 in tumor sites means a lack of metastasis and tumor recurrence with a consequent better prognosis. The accumulation of T cells CD3+CD8+ carries out a lack of metastasis [33].

#### 3.2.36. Tregs

T regulatory cells are IL-17+ Foxp3+ CD4+ T cells with high expression of IL-2R (CD25). TReg has a pleiotropic function in relation to the expression of the FOXP3 box, in particular, in patients with CAC where it is correlated either with short disease-free survival or with reduced tumor growth. Its primary role is in the maintenance of homeostasis and preventing activation against self-antigens or harmless antigens; in this case, the activation of the FoxP3 box is temporary. TRegs accumulate in TME in case of cytokine and chemokine signaling, which is produced by dysplastic cells and stromal fibroblasts, or via CD14+ macrophages and CD11+ DC (also in an independent manner from cytokine signaling [34]), with the aim to suppress the anti-tumorigenic activity of CD8+ T cells and NK. In this latter case, the expression of the FoxP3 box is permanent. In addition, TRegs activated in the tumor context are extremely respondent to low concentrations of their main activator, IL-2, due to the acidity of CD25. This can lead to two situations: reduction of levels of available IL-2 and, consequently, enhanced TReg-mediated suppression activity. TRegs act in two main ways: in granzyme and perforin, they physically destroy cell membranes and inactivate the extracellular conversion of ATP in adenosine. In CAC, the role of TRegs may vary in function of their localization. At the tumor periphery, TRegs reduce the activity of the immune system, fighting against tumor growth. In the tumor core, it suppresses pro-inflammatory and pro-tumorigenic activity [32]; an interesting aspect is that they are observed only in tumor foci, not in adjacent tissues. In CAC, besides CD25, TRegs express RORγt, giving them the ability to secrete cytokines, such as IL-17, thereby worsening the clinical outcome while the deficiency of RORγt protects against polyposis and restores the antitumor actions to TRegs [34]. In the late stages of CAC, a number of TRegs increasingly colonize mesenteric lymph nodes [32]. CD39 + γδ TRegs act as suppressors of T cell activity more than other Tregs, and their expression depends on TGFβ1 [35].

#### 3.2.37. IL-35

IL-35 promotes T cell exhaustion in TME [32].

#### 3.2.38. IL-32

IL-32 plays different roles in the function of different isoforms. IL-32γ enhances the anti-cancer activity of TNF-α and blocks the NF-kB-STAT3 pathway. IL-32θ directly inhibits STAT3 with the purpose of inhibiting cell transition from epithelial to mesenchymal. IL-32θ levels negatively correlate with the grade of tumor, and it is expressed only in the first steps of tumorigenesis [28].

#### 3.2.39. S1P

Sphingosine-1-phosphate is necessary for activation of NF-kB TNFα-dependent and IL-6 NF-kB-regulated production. The presence of the S1P receptor is induced by STAT3, which in turn, is necessary for persistent activation of STAT3 [36].

#### 3.2.40. CD73

Hypoxia, HIF1, TGF-β, IL-6, IFNs, TNF-α, IL-1β enhance the expression of CD39 and CD73, which lead to the accumulation of adenosine in TME. CD73 is expressed on the cell surface of B cells, T cells, MDSCs, Tregs and M2 macrophages, inhibiting their activity and promoting T cell infiltration of the tumor site. Its increasing expression is directly correlated with tumor progression and poor outcome. It is related to the expression of β-catenin and increases the expression of EGFR [35].

#### 3.2.41. Adenosine/Adenosine Receptors

Low levels of adenosine in TME stimulate the growth of tumor cells, while high levels carry out immunosuppression. Adenosine receptors 1 and 2 have an apoptotic role, receptor 3 oversees cell proliferation by activating c-myc and the ERK pathway [35].

#### 3.2.42. MIF

Macrophage migration inhibitory factor belongs to a pro-inflammatory cytokine and is secreted from macrophages, myofibroblasts and T cells under regulatory stimulus from IFNγ, TGFα, TGFβ, IL-1β and IL-25. MIF acts as an immobilizer of macrophages in the tumor site, in transition from epithelial to mesenchymal cells always in the tumor site and reversed in the bloodstream to allow the implant of metastases. MIF has an impact on Akt signaling, production of EGFR and activation of Erk [37].

#### 3.2.43. MAPK2

Responsible for post-transcriptional enzymatic modification of the molecular structure of IL-1β, IL-6 and TNF-α. Their production decreases approximately 80% in the case of MAPK2 silencing either in tumor cells or immune cells; the number of tumor macrophages decreases as well [38].

#### 3.2.44. Gankyrin

Gankyrin is an oncoprotein contained in myeloid cells. Its presence is positively linked to IL-17 levels, while its reduction determines a stop of STAT3/MAPK, which means a reduction of levels of TNFα [39].

Figure 2 depicts the main molecules in the pathways described in the text.

## 4. Discussion

A strong link between inflammation and cancerogenesis was already presumed at the time of Hippocrates (460 B.C) and Galenus (129 A.D.) [40]. In 1863, Virchow added the “chronic” adjective to that link, described by Greek physicians, and was the first in medical history who discussed a lymphoreticular infiltrate in the tumor site [41,42]. Almost 30 years later, starting from the assumption that some microbial products could have anti-tumoral effects, W.B. Coley provided injections to his oncologic patients of a mix of bacterial toxins, observing an effective improvement of clinical conditions [43,44]. In 1925, B. Crohn, de facto paving the way for the study of chronic inflammatory diseases, was the first to demonstrate the tight link between inflammation and colon cancer [45]. Sixty years later, a group of the Mount Sinai Hospital of New York published the first study that showed that people with IBD, including Crohn’s, almost certainly have an increased risk of colon cancer [46]. The last great milestone in the field of inflammation and carcinoma was posed in 2000 when Hanahan and Weinberg drew up their famous model with six hallmarks of tumorigenesis: self-sufficiency in growth signals, insensitivity to anti-growth signals, evading apoptosis, limitless replicative potential, sustained angiogenesis and tissue invasion and metastasis [47]. A seventh was added later: inflammation [48].

In fact, as seen in the Results, inflammation is a starter and trigger in the tumorigenesis process, modulating the immune system and stimulating neoplastic and non-neoplastic cell growth, local invasion and metastatic diffusion, as shown in Figure 3 [47,49].

Unlike what happens in sporadic carcinomas, tumorigenesis driven by inflammation shows several peculiarities.

In sporadic colon cancer tumorigenesis, the chronological sequence is loss of APC, aneuploidy, methylation, microsatellite instability, activation of K-Ras and COX-2, change in DCC/DPC4 genes and loss of function of p52 [50]. In inflammation-driven colon carcinomas, the sequence is subverted—mutation of p53 and k-RAS occur earlier because the β-catenin pathway does not need a genetic mutation to start [51,52].

Furthermore, as a result of the production of various circulating factors and genetic modifications, chronic inflammation of the colon could predispose to other malignancies causing not only local effects but also systemic ones [53,54].

In clinical practice, risk to develop colonic cancer in patients affected by IBD is time-dependent: 1.6% after 10 years of IBD, 8.3% after 20 years, 18.4% after 30 years. According to a metanalysis by Eaden et al., IBD patients hold an overall risk of developing colon cancer of 3.7% compared with the normal population [55]. According to Rutter et al., that risk decreases to 2.5% in 20 years, 7.6% in 30 years and 10.8% in 40 years [56]. Twenty-one percent of IBD patients develop colon cancer within 10 years of being diagnosed with inflammatory disease (estimated median time of 17 years) [57].

Four risk factors for developing carcinoma in chronic inflammation conditions are set out below:Longitudinal extension of inflammation is an independent risk factor. An estimated risk of 5.4% for pancolitis, moderate risk in left-sided inflammation and low risk (almost the same as non-affected population) in cases of proctitis are reported [58,59,60].Age of onset is closely associated with the development of cancer (estimated risk of 43.8% in ages from 0 to 19 years old and 2.65% in a range from 20 to 39 years old) [61,62,63].Primary sclerosing cholangitis (PSC) is a predisposing condition for the development of colon cancer and cholangiocarcinoma. Most authors reported an estimated risk of 9% after 10 years, 31% after 20 years, 50% after 25 years, probably due to an abnormal exposition of the mucosal and submucosal layers to biliary products.The risk of developing bowel cancer may be higher in the case of a family history of the disease (risk factor independent of the overall burden of the disease or the age of the onset) [63,64].

European Crohn’s and Colitis Organization, American Gastroenterological Association, British Society of Gastroenterology and American Society for Gastrointestinal Endoscopy from 2010 to 2013 made a strategy of surveillance in six points: colonoscopy in remission phases of IBD, first colonoscopy after 8–10 years from IBD diagnosis, annual colonoscopy, concomitant PSC requires annual colonoscopy since diagnosis, systematic bioptic procedure during colonoscopy is necessary (bioptic sample every 10 cm of exploration), dysplasia findings on biopsy deserve an anatomopathological second-look [65].

Despite this, a high percentage (50–80%) of misdiagnosed colon carcinoma lesions were found during surveillance colonoscopies in IBD patients [60]. The present paper has two main limitations: the first one is the exclusion from the first identification of all papers present in registries. This choice was due to the extremely large number of records found in databases. The second one was the difficulty of finding the same quantity of information for each molecule/cell that participates in carcinogenesis. In fact, IL-6 and IL-27 are heavily studied, and their pathways are well understood. However, there are molecules such as IL-35 where the available information is too limited.

The study protocol was made with the aim of reducing, at much as possible, the risk of bias. For this reason, the group of researchers was divided into two subunits who worked independently for the first part of the paper with meetings only in the final steps of paper production with the academic author who acted as moderator. Clearly, the risk of bias is not avoidable, and here, there were two possible biases: the first one is the scoring system that was set to assign a measurable number for uncountable items such as “quality of title” to uniform the judgment of authors; the second one is the usage of strict exclusion criteria, which resulted in a high number of excluded papers.

The present paper could be considered a resume of what is known about each molecule/cell involved in the transition from inflammation to CAC. Many molecules, such as 5-ASA, are under a magnifying glass due to their property to interfere with some carcinogenic pathways, but there are many other nodes in the pathways that could be used as a target for pharmacological treatment. This is desirable in early diagnosis to use an evolution of new techniques in endoscopic or radiological fields and, as a consequence, provide a more focused surgical treatment.

## 5. Other Information

The present paper was registered in the OpenScience Framework registry (Identifier: DOI 10.17605/OSF.IO/2KG7T). Internet Archive link https://archive.org/details/osf-registrations-b6fg3-v1 (accessed on 10 December 2021).

## 6. Conclusions

Despite the acquired information in the treatment of colon cancers, the team of authors of this paper experiment every day with differences in diagnosis, treatment and outcome of sporadic CRC and CAC. It is well known that tumorigenesis step way is completely different among the two pathologies, and this is noted every time that a patient with IBD is hospitalized or arrives for an outpatient consultation.

As demonstrated, the increased cancer risk can have a real sword of Damocles upon the heads of patients with IBD, even if the immune system and inflammation do not act alone in cancer development, progression and diffusion.

Firstly, the immune system acts as a pro-tumorigenic but also as an anti-tumorigenic. Assuming that inflammation is not always a clinical foe, during the different stages of tumorigenesis, the immunity system can prevent and fight tumor growth.

Secondly, trying to know every single molecule and pathway is a real challenge due to the thick network of connections among them. This may lead to difficulties in pharmacological treatments caused by the possibility of blocking one pathway while another one escapes, from the extent of treatments and in surgical treatments caused by the possibility of high recurrences in post-op patients.

The spirit that animated the group of authors was to try to sum up in one review most of what is known about IBD and CAC with the first intent to comprehend what is experienced day by day at the patient’s bed firsthand. The second intent is that this review could be a new tool on the way to discovering, in a few years, new pharmacological molecules that could act in enhancing or suppressing pathways, new screening protocols for patients with IBD and new surgical weapons against CAC or pre-carcinogenicity conditions.

## Figures and Tables

**Figure 1 cancers-13-06237-f001:**
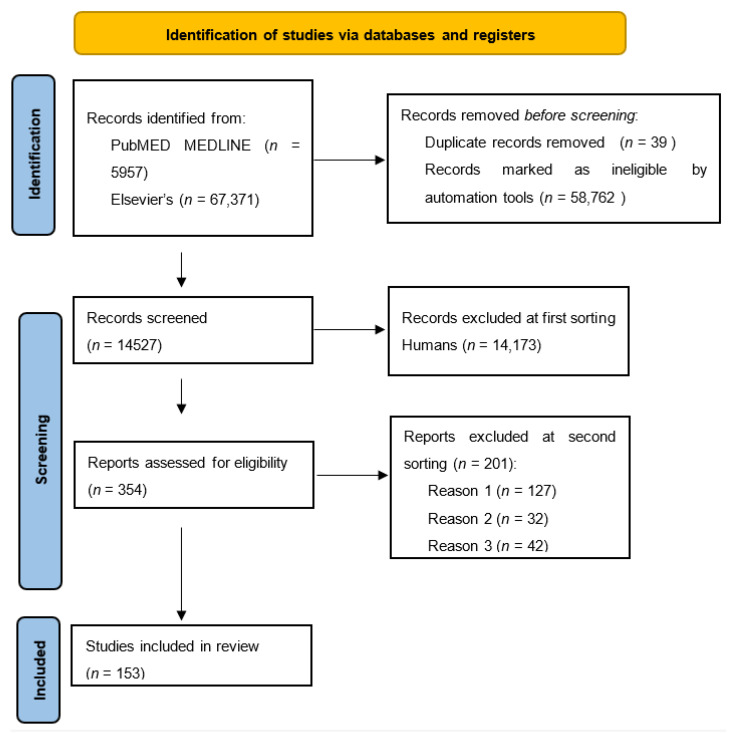
PRISMA^®^ Flow diagram for new systematic reviews, which included searches of databases and registers [3].

**Figure 2 cancers-13-06237-f002:**
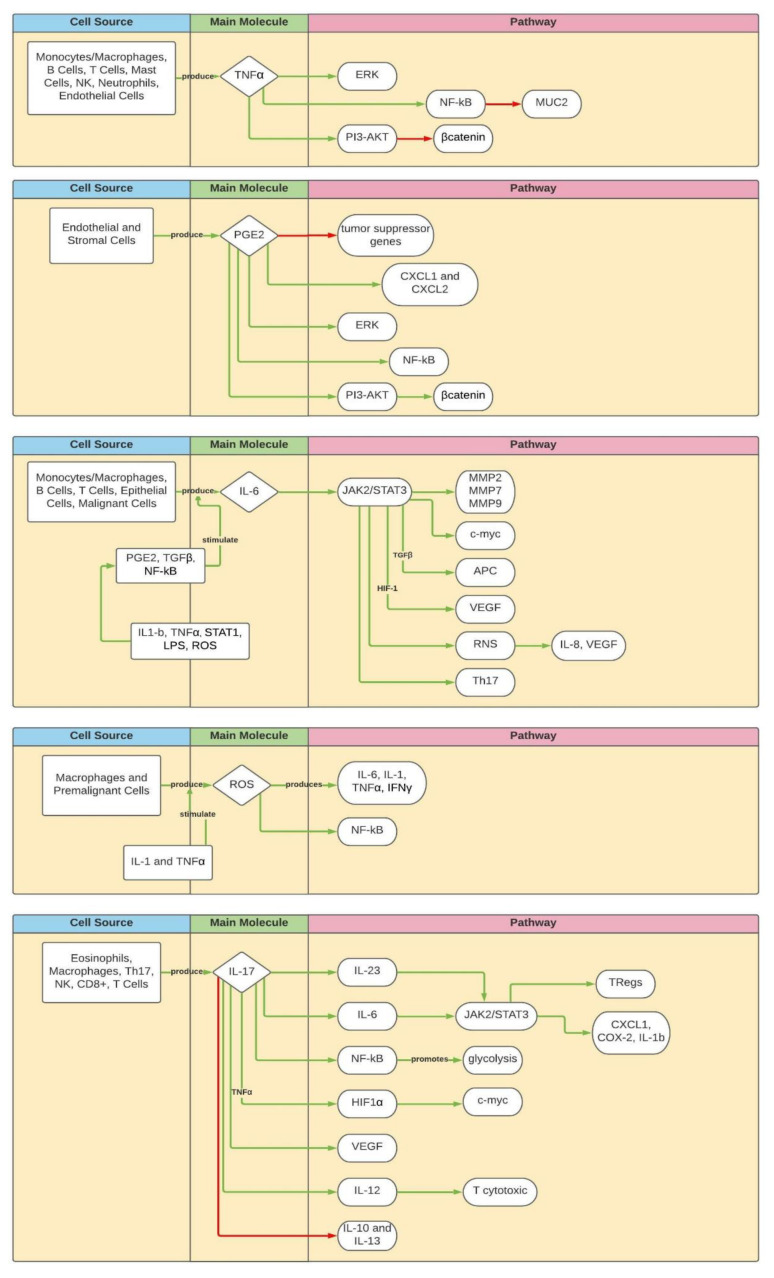
Principal pathways. Green arrows: activating/enhancing action. Red arrows: deactivating/suppressing action.

**Figure 3 cancers-13-06237-f003:**
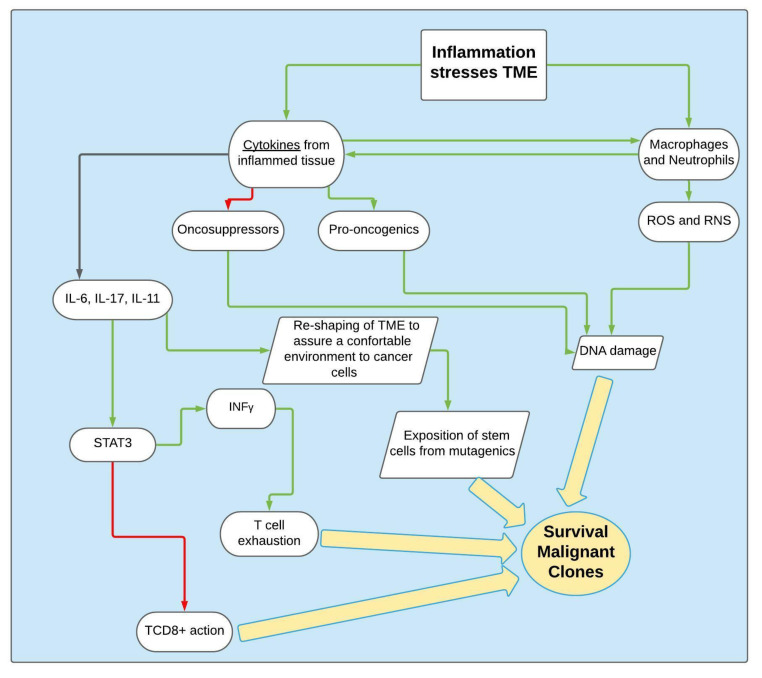
How inflammation induces CAC. Green arrows: activating/enhancing action. Red arrows: deactivating/suppressing action.

**Table 1 cancers-13-06237-t001:** Scoring system.

Sorting	Analyzed Part of the Paper	Parameter	Judgment
First Sorting	Quality of title	Relevance	1–5
	Quality of abstract	Clarity of objectives	1–5
		Definition of project	1–5
		Clarity of exposition	1–5
	Quality of keywords	Relevance	1–5
	Result for first sorting		Included Uncertain Excluded
Second Sorting	Quality of full text	Clarity in study design	1–5
		Characteristics of population	Homogeneous Inhomogeneous
		Size of population	Not Representative Low Representative Representative High Representative
		Analytic procedures	Declared Not declared
		Bias	Declared Not declared
		Intracellular pathways	Yes No
	Final result		Included Excluded

**Table 2 cancers-13-06237-t002:** Classification of actors of inflammation according to their activity.

Actors of Inflammation
Pro-Tumorigenic	Anti-Tumorigenic	Hybrid Action
IL-6	TLR	I-kB	IL-22
IL-11	Stromal cells	COX-1	MSC
IL-23	VEGF	IL-37	TGF-β
IFN-γ	IL-8	IL-21	IL-10
NF-Kb	IL-1β		T cells
ROS	STAT3		TRegs
RNS	NOD2		IL-32
TNFα	Βcatenin (Wnt/Βcatenin)		Adenosine Receptor 1 and 2
COX-2	IL-35		MAPK
PGE2	S1P		Gankirin
MDSCs	CD73		IL-17
IL-1	Adenosine receptor 3		
Th-17	MIF		
CXCL8	CCL20		
CCL5	CCL2

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
