# Peer review of "The Complex Network between Inflammation and Colorectal Cancer: A Systematic Review of the Literature"

_cancers, 2021, doi:10.3390/cancers13246237_

Round 1
Reviewer 1 Report
#The authors of the review have compiled the review of literature of importance and significance to researchers of the field.
#The overall review is impressive but requires english language corrections.
#The introduction can be made more extensive including the importance of colorectal cancer and the relevance of the present review.
#The authors have included detailed molecular targets like IL-6,NFKB,COX1 and 2 etc which is commendable.
#The figures included in the review #2 and #3 can be enlarged a bit or the text more readable.
Author Response
COVER LETTER – REVISED MANUSCRIPT ID: Cancers-1465186
November 22, 2021
Please find enclosed our revised manuscript entitled “The complex network between inflammation and colorectal cancer: a systematic review of the literature”.
As you can read in the paper, we are a group of surgeons with expertise in colorectal cancer treatment who made a little trespass into preclinical field whit the aim to better comprehend pathogenetic mechanisms and to try to find an explication to the different natural history of Colitis-Associated Cancer, when compared with sporadic colorectal cancer. To attempt to this objective, in this paper we made a sum of reviews available, who subjected the strict link between inflammation and colorectal cancer.
We appreciate very much the reviewers’ approval and we worked to make the modifications suggested by them and by editor.
You can easily recognize all modifications on the original text because we made them by using the “Track Changes” tool of Microsoft World program (all modifications are in red font).
Editor’s requests
- Institutional email addresses of the authors:
Author |
Institutional mail |
Dr. Rossana Percario |
rossana.percario@asl2abruzzo.it |
Dr. Paolo Panaccio |
paolo.panaccio@asl2abruzzo.it |
Dr. Fabio Francesco di Mola |
fabio.dimola@grupposynergo.com |
Dr. Tommaso Grottola |
tommaso.grottola@grupposynergo.com |
Prof. Dr. Pierluigi di Sebastiano |
pierluigi.disebastiano@unich.it |
- The copyright was registered at link (https://creativecommons.org/licenses/by/4.0/). Here you are the screenshot as a proof of registration:
The Rich Text into the red dotted outline box was copied and pasted directly in the paper, in the left-sided column at first page.
- The list of reference was reformatted.
- PRISMA checklist was added to the paper as Table 1 and contextualized into the text.
Replies to the Reviewer #1:
#2. About English correction, authors would request the English revision service provided by the Editor.
#3. The introduction was modified according with reviewer suggestions.
#5. Figures 2 and 3 were completely re-edited enlarging the text box and augmenting the size of the font.
Replies to the Reviewer #2:
#1. The paper was enriched with Table 2 which is a synthesis of the tools that we used to choose the good papers for our review.
#2. In sorting phase1 and 2, we calculated the relevance of each single paper examined, not the relevance of the molecule. The principle was: a good concepted paper who meets the criteria declared in our tool-box, this will be more reliable under scientific profile than a simple “copy/paste” paper. For this reason, a high number of papers was excluded. From each admitted paper, the molecule or pathway was extrapolated and inserted into the paragraph 3.2.
#3. The figure 3 was modified
#4. The conclusion was modified.
We believe our reviewed manuscript would meets the expectation of Cancers editor and reviewers.
Thank you for your consideration of our work. Please address all correspondence to the first author rox.percario@live.it and to academic author diseb@hotmail.com.
Sincerely,
Dr. Rossana Percario and Prof. Dr. Pierluigi di Sebastiano

Reviewer 2 Report
Percario and colleagues provide a systematic review of the literature on the complex networks proposed to connect inflammation and colo-rectal cancer. This is not the first review on the topic but the attempt to extract a message from the data present in the literature is well taken and the straightforward approach is very much appreciated. However, the manuscript needs improvement to convey a clearer message. Hereunder are some suggestions.
- This reviewer would request the authors to include a table summarizing the parameters considered and the scoring used, which is explained in paragraph 1 of methods (line 63 to 70). A table would help the reader and deliver a stronger message.
- For those who are not familiar with the PRISMA tool, it is not clear how the relevance of all the inflammatory mediators described in paragraph 3.2 was calculated after the sorting of the literature and the selection of 150 papers. Also, it is not clear whether the information related to every single mediator were taken from the 150 papers or others. In general, what is missing is a paragraph describing in more detail the steps taken to perform the analysis of the 150 papers.
- In Figure 2. Unless there is a precise nomenclature related to these graphs, I would substitute Progenitors with Cell type or Cell source.
- The Conclusion paragraph should be rephrased to convey a clearer message.
Minor points:
Line 73. Aiming of reducing bias of selection.. would read better Aiming at reducing bias of selection.
Line 79. All the articles were (not) are deliberately excluded..
Line 103. Determine (not) determines.
Line 375. Linked (not) lied.
Author Response

(The authors gave the same response as above.)

Round 2
Reviewer 2 Report
In order to address one of my points, authors included the PRISMA checklist table, which can be downloaded from the internet
http://prisma-statement.org/PRISMAstatement/checklist.aspx
I am not sure this is allowed, Authors may consider to modify the format of the table, without affecting the content.
Author Response
PRISMA checklist was added in first revision phase under indication of the Assistant Editor. Talking with the Managing Editor, she told us that the checklist must be enclose at submission as a separated file but not into the main text. By considering that, we removed completely the table and the relative contextualization.